# MGIC: Multigrid-in-Channels Neural Network Architectures

**Moshe Eliasof** [*]    **Jonathan Ephrath** [*]    **Lars Ruthotto** [†]    **Eran Treister** [*]

## Abstract

Multigrid (MG) methods are effective at solving numerical PDEs in linear complexity. In this work we present a multigrid-in-channels (MGIC) approach that tackles the quadratic growth of the number of parameters with respect to the number of channels in standard convolutional neural networks (CNNs). Indeed, lightweight CNNs can achieve comparable accuracy to standard CNNs with fewer parameters; however, the number of weights still scales quadratically with the CNN's width. Our MGIC architectures replace each CNN block with an MGIC counterpart that utilizes a hierarchy of nested grouped convolutions of small group size to address this. Hence, our proposed architectures scale linearly with respect to the network's width while retaining full coupling of the channels as in standard CNNs. Our extensive experiments on image classification, segmentation, and point cloud classification show that applying this strategy to different architectures reduces the number of parameters while obtaining similar or better accuracy.

## 1 Introduction

Convolutional neural networks (CNNs) [23] have achieved impressive accuracy for many imaging tasks [22, 11], and numerical PDE solvers [20, 2]. The main idea behind CNNs is to define the linear operators in the neural network as convolutions with local kernels. This increases the network's computational efficiency (compared to the original class of networks) due to the compact convolution operators and the considerable reduction in the number of weights. The general trend in the development of CNNs has been to make deeper and wider networks to achieve higher accuracy [35].

In practical applications of CNNs, a network's feature maps are divided into channels, and the number of channels, $c$, can be defined as the width of the layer. A standard CNN layer connects any input channel with any output channel. Hence, the number of convolution kernels per layer is equal to the product of the number of input channels and output channels. Assuming the number of output channels is proportional to the number of input channels, this $\mathcal{O}(c^2)$ growth of operations and parameters causes immense computational challenges. When the number of channels is large, convolutions are the most computationally expensive part of the training and inference of CNNs.

Wide architectures exacerbate this trend with hundreds or thousands of channels, which are particularly effective in classification tasks involving a large number of classes. Increasing the network's width is advantageous in terms of accuracy and computational efficiency compared to deeper, narrower networks [48]. However, the quadratic scaling causes the number of weights to reach hundreds of millions and beyond [18], and the computational resources (power and memory) needed for training and running such CNNs surpasses the resources of common systems [4]. This motivates us to follow multigrid approaches that can solve numerical PDEs in linear complexity by utilizing a hierarchy of grids. Using the same approach, we aim to design more efficient network architectures with competitive performance.

[*]Computer-Science Department, Ben-Gurion University of the Negev. (eliasof, ephrathj@post.bgu.ac.il, erant@cs.bgu.ac.il)

[†]Departments of Mathematics and Computer Science, Emory University. (lruthotto@emory.edu)

35th Conference on Neural Information Processing Systems (NeurIPS 2021), Sydney, Australia.

**Algorithm 1** Multigrid-in-channels block

---

$\mathbf{x}_{l+1} = \mathbf{MGIC\text{-}block}(\mathbf{x_l}, \text{CNN-block}, s_g, s_c)$.
*# $\mathbf{x}_l$ - input feature map with $c_{\text{in}}$ channels.*
*# $s_g$: group size. $s_c$: coarsest grid size.*
*# CNN-block: A reference CNN block.*
$\mathbf{x}^{(0)} = \mathbf{x}_l \quad n_{\text{levels}} = \left\lfloor \log_2\left(\frac{c_{\text{in}}}{s_c}\right) \right\rfloor$
**for** $j = 0 : n_{\text{levels}}$ **do**
$\quad \Big| \quad \mathbf{x}^{(j+1)} = R_j \mathbf{x}^{(j)}$
**end**
$\mathbf{x}^{(n_{\text{levels}})} \leftarrow \text{CNN-block}(\mathbf{x}^{(n_{\text{levels}})})$
**for** $j = n_{\text{levels}} - 1 : 0$ **do**
$\quad \Big| \quad \mathbf{x}^{(j)} \leftarrow \mathbf{x}^{(j)} + \mathcal{N}\left(P_j(\mathbf{x}^{(j+1)} - R_j \mathbf{x}^{(j)})\right)$
$\quad \Big| \quad \mathbf{x}^{(j)} \leftarrow \text{CNN-block}(\mathbf{x}^{(j)}, \text{group\_size} = s_g)$
**end**
return $\mathbf{x}_{l+1} = \mathbf{x}^{(0)}$.

---

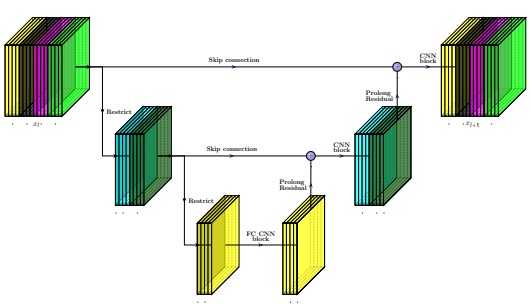

Figure 1: A three-level multigrid block for 16 input channels and a group size of four. *Restrict* and *Prolong Residual* denote grid transfer operators, which decrease and increase the number of channels, respectively. All the channels in the block are of the same spatial resolution. Each color denotes a group of channels that are mixed in a CNN block. The coarsest level uses a FC CNN block.

## 2 Multigrid-in-channels CNN architectures

Typical CNN architectures are composed of a series of blocks

$$\mathbf{x}_{l+1} = \text{CNN-block}(\mathbf{x}_l), \tag{1}$$

where $\mathbf{x}_l$ and $\mathbf{x}_{l+1}$ are the input and output features of the $l$-th block, respectively. Each CNN-block usually contains a sequence of basic layers, with associated weights which are omitted in the following. A convolution layer takes $c_{\text{in}}$ channels of feature maps, and outputs $c_{\text{out}}$ such channels, meaning $\mathcal{O}(c_{\text{in}} \cdot c_{\text{out}})$ parameters and FLOPs. We propose to replace the CNN-block in (1) by a novel multigrid block to obtain the forward propagation

$$\mathbf{x}_{l+1} = \text{MGIC-block}(\mathbf{x}_l, \text{CNN-block}, s_g, s_c), \tag{2}$$

which, as illustrated in Fig. 1, uses a hierarchy of grids in the channel space and applies the original CNN-block on the coarsest level. The parameter $s_g$ defines the group size of the convolution operators in these CNN-blocks, and $s_c$ is the size of the coarsest grid. As we show in Sec. A, the number of parameters and FLOPs in the MGIC-block scale *linearly* with respect to the number of channels, assuming that the group size is fixed. Note that the MGIC block is agnostic to its CNN-block, and therefore can be used for various CNN architectures, including future ones.

### 2.1 The multigrid hierarchy

We design a hierarchy of grids in the channel space (also referred to as "levels"), where the number of channels in the finest level corresponds to the original width of the network. The number of channels is halved between the levels until reaching the coarsest level, where the number of channels is smaller or equal to the parameter $s_c$. Our multigrid architecture is accompanied by a CNN block, like a ResNet block [13] which is applied on each level. On the finest and intermediate levels, we only connect disjoint groups of channels using grouped convolutions. These convolutions have $\mathcal{O}(s_g \cdot c_{\text{in}})$ parameters, and we keep the group size $s_g$ fixed throughout the network. Hence, as the network widens, the number of groups grows, and the number of parameters grows linearly. We allow interactions between all the channels on the coarsest grid, where we use the original CNN-block without grouping. Hence, that our architecture performs more convolution layers and non-linear activations per MGIC-block, which is designed to replace a given CNN-block, yielding higher capacity and expressiveness at similar computational cost.

### 2.2 The multigrid block

Assume that the CNN-block and the MGIC-block change neither the number of channels nor the spatial resolution of the images. That is, both $\mathbf{x}_l$ and $\mathbf{x}_{l+1}$ in (2) have $c_{\text{in}}$ channels of the same spatial resolution. Given a CNN-block, a group size $s_g$ and a coarsest grid size $s_c$, we define the multigrid

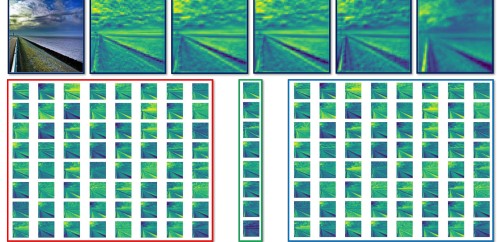

| $s_g$ | 32 | 16 | 8 | 4 |
|---|---|---|---|---|
| MSE ($10^{-3}$) | 11 | 13 | 17 | 24 |
| Params [K] | 3.3 | 1.8 | 0.9 | 0.4 |

Table 1: Feature maps reconstruction MSE v.s. $s_g$. $s_c$ is fixed to 8.

Figure 2: Left to right: Top - input image, a feature map, and its reconstructions with $s_g = 32, 16, 8, 4$, respectively ($s_c = 8$). Bottom - input feature maps, their coarsest grid representation, and their MGIC reconstruction.

block in Alg. 1, and as an example we present a two-level hierarchy denoted by levels $0, 1$, where $\mathbf{x}^{(0)} = \mathbf{x}_l$ are the input feature maps at the finest level (level 0). The two-level block is as follows

$$\mathbf{x}^{(1)} \quad = \quad R_0 \mathbf{x}^{(0)} \tag{3}$$

$$\mathbf{x}^{(1)} \quad \leftarrow \quad \text{CNN-block}(\mathbf{x}^{(1)}) \tag{4}$$

$$\mathbf{x}^{(0)} \quad \leftarrow \quad \mathbf{x}^{(0)} + \mathcal{N}(P_0(\mathbf{x}^{(1)} - R_0 \mathbf{x}^{(0)})) \tag{5}$$

$$\mathbf{x}_{l+1} \quad = \quad \text{CNN-block}(\mathbf{x}^{(0)}, s_g) \tag{6}$$

We first down-sample the channel dimension of the input feature maps $\mathbf{x}^{(0)}$ in Eq. (3) by a factor of 2, using a restriction operator $R_0$. This operation creates the coarse feature maps $\mathbf{x}^{(1)}$, which have the same spatial resolution as $\mathbf{x}^{(0)}$, but half the channels. The operator $R_0$ is implemented by a grouped $1 \times 1$ convolution; see a detailed discussion in Appendix A. Then, in Eq. (4) a non-grouped CNN block is applied on the coarse feature maps $\mathbf{x}^{(1)}$. Following that, in Eq. (5) we use a prolongation operator $P_0$ to up-sample the residual $\mathbf{x}^{(1)} - R_0 \mathbf{x}^{(0)}$ from the coarse level to the fine level (up-sampling in channel space) and obtain a tensor with $c_{\text{in}}$ channels. Finally, in Eq. (6) we perform a grouped CNN block. An illustration of this architecture using three levels is presented in Fig. 1. The multilevel block is applied to reduce the channel dimension to the coarsest grid size $s_c$.

## 3 Experiments

In this section, we report several experiments with our MGIC approach. Additional experiments and details are given in Appendix B.

**Coarse channels representation** We wish to quantify the effectiveness of our channel down and up sampling mechanism by measuring how well we can encode feature maps on the coarsest grid. We sample $1,024$ images from ImageNet, and extract their feature maps from the first convolution layer of a pre-trained ResNet-50 with 64 channels. Then, we encode and decode the feature maps using the restriction and prolongation operators, respectively. To study the transfer operators in isolation, we remove the CNN blocks and long skip connections in Fig. 1 from the MGIC-block. We experiment with several values of the group size parameter $s_g$ and present the mean squared error of the feature maps reconstruction in Tab. 1. The original feature maps and their reconstructions in Fig. 2. According to this experiment, our method is capable of faithfully representing the original channel space (obtaining low MSE values).

**Performance in function approximation** We claim that if a network can faithfully approximate a function $f(\vec{x}) : \mathbb{R}^n \to \mathbb{R}$, then its capacity, or representation power is high. This property is important, especially in applications where we wish to model implicit function via a neural networks, such as signed-distance fields for shape reconstruction and completion [31], and solution of PDEs [30, 33, 3, 26]. Here, we approximate the function $f(x, y) = \cos(x)\sin(20y)$, given $(x, y) \in [0, 1]^2$. We use three networks – MobileNetV3 [16], GhostNet [12] and our MGIC. The results are given in Appendix B.1 suggesting that our MGIC has higher capacity, as it yields lower MSE with a lower number of parameters.

| Model | Params [M] | FLOPs [M] | Top-1 Acc.% |
|---|---|---|---|
| MNetV3-S 0.75× | 2.4 | 44 | 65.4 |
| GhostNet 0.5× | 2.6 | 42 | 66.2 |
| MGIC 0.6× | 2.3 | 48 | **67.0** |
| MNetV3-L 0.75× | 4.0 | 155 | 73.3 |
| GhostNet 1.0× | 5.2 | 141 | 73.9 |
| MGIC 1.0× | 5.2 | 145 | **74.8** |
| MNetV3-L 1.0× | 5.4 | 219 | 75.2 |
| GhostNet 1.3× | 7.3 | 226 | 75.7 |
| MGIC 1.2× | 7.1 | 233 | **76.1** |

Table 2: Comparison of light-weight networks on ImageNet dataset classification. MNetV3 denotes MobileNetV3, and our MGIC operates with MobileNetV3 as a CNN-block.

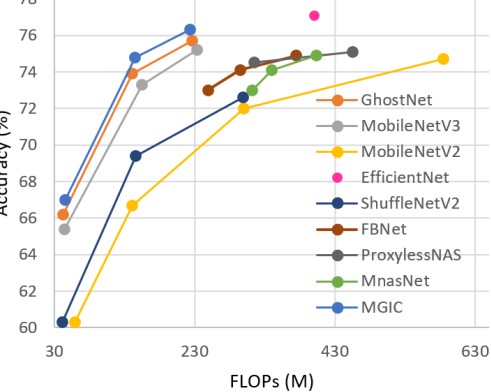

Figure 3: Accuracy on ImageNet.

**ImageNet classification on a budget of FLOPs** The ImageNet [9] challenge ILSVRC 2012 consists of over $1.28M$ training images and $50K$ validation images from $1000$ categories. We use SGD optimizer with a mini-batch size of $256$ for $100$ epochs with cross-entropy loss. The initial learning rate is $0.1$, divided by $10$ every $30$ epochs. The weight decay is $0.0001$, and the momentum is $0.9$. As data augmentation, for both datasets, we resize the images to $224 \times 224$ and use standard random horizontal flipping and crops, as in [13]. We follow the MobileNetV3-Large [16] architecture for its efficiency and high accuracy, and replace the standard MobileNetV3 block with our MGIC block, configured to $s_g = 64, s_c = 64$. Three scales of networks are reported with width factors of $0.6, 1.0$ and $1.2$, respectively. We find that our method obtains higher accuracy, with a similar number of FLOPs. We present our results in Fig. 3 and Tab. 2.

**Point cloud classification** The previous experiments were performed on structured CNNs, i.e., on 2D images. To further validate our method's generalization and usefulness, we incorporate it in graph convolutional networks (GCNs) to perform point cloud classification. Specifically, we use a smaller version of the architecture from [39], where we alter the width of the last three classifier layers from $1024, 512, 256$ to $64$ in all of them, as described in Appendix B.4. Table 3 summarizes the results

Table 3: ModelNet-10 classification.

| Backbone | Params[M] | FLOPs [M] | Accuracy % |
|---|---|---|---|
| DGCNN [39] | 0.16 | 125 | 91.6 |
| diffGCN [10] | 0.57 | 64 | 92.5 |
| **MGIC-diffGCN (ours)** | **0.11** | **13.7** | **92.9** |

## 4    Conclusion

We present a novel multigrid-in-channels (MGIC) approach that improves the efficiency of convolutional neural networks (CNN) both in parameters and FLOPs, while using easy-to-implement structured grouped convolutions in the channel space. Applying MGIC, we achieve full coupling through a multilevel hierarchy of the channels, at only $\mathcal{O}(c)$ cost, unlike standard convolution layers that require $\mathcal{O}(c^2)$. This property is significant and desired both to reduce training and inference times, which also translates to a reduction in energy consumption. We also note that MGIC is most beneficial for wide networks, which are usually favored for state-of-the-art accuracy and performance. Our experiments for various tasks suggest that MGIC achieves comparable or superior accuracy than other recent light-weight architectures at a given budget. Our MGIC block offers a universal approach for producing lightweight versions of networks suitable for different kinds of CNNs, GCNs, and traditional NNs, where fully-connected layers are applied. Furthermore, it is future-ready, meaning it can also compress future architectures when available.

## Acknowledgments

The research reported in this paper was supported by the Israel Innovation Authority through Avatar consortium, and by grant no. 2018209 from the United States - Israel Binational Science Foundation (BSF), Jerusalem, Israel. ME is supported by Kreitman High-tech scholarship.

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

## A  Multigrid-in-channels CNN architectures

In this following section, we provide details about the design and implementation of our method.

**The choice of transfer operators $P$ and $R$**

The transfer operators play an important role in multigrid methods. In classical methods, the restriction $R$ maps the fine-level state of the iterative solution onto the coarse grid, and the prolongation $P$ acts in the opposite direction, interpolating the coarse solution back to the fine grid. Clearly, in the coarsening process we lose information, since we reduce the dimension of the problem and the state of the iterate. The key idea is to design $P$ and $R$ such that the coarse problem captures the subspace that is causing the fine-grid process to be inefficient. This results in two complementary processes: the fine-level steps (dubbed as *relaxations* in multigrid literature), and the coarse grid correction.

To keep the computations low, at the $j$-th level we choose $R_j$ to be a grouped $1 \times 1$ convolution that halves the number of channels of its operand. We choose $P_j$ to have the transposed structure of $R_j$. For $R_j$ and $P_j$ we choose the same number of groups as in the CNN-block , e.g., for $R_0$ it will be $\frac{c_{\text{in}}}{s_g}$ groups. The operators $R_j$ take groups of channels of size $s_g$ and using a $1 \times 1$ convolution distills the information to $\frac{s_g}{2}$ channels. Similarly, the operators $P_j$ interpolate the coarse channels back to the higher channel dimension using grouped $1 \times 1$ convolutions. This choice for the transfer operators corresponds to aggregation-based multigrid coarsening [37], where aggregates (groups) of variables are averaged to form a coarse grid variable. In Sec. 3, we exemplify that the transfer operators preserve essential information of all channels.

The weights of the transfer operators are learned as part of the optimization, and are initialized by positive weights with row-sums of 1. The purpose of this initialization is that the feature maps will not vanish as we multiply them by consecutive restrictions $R_j$ to start the MGIC-block.

**The importance of up-sampled residuals**

Adding the up-sampled residual in (5) is the standard way to apply multigrid methods to solve non-linear problems [38, 45]. Here, it allows us to have a skip connection between corresponding levels of the multigrid, introducing an identity mapping in (5) as guided by [13]. To prevent exploding gradients by feature maps summation, at each level $j$ we up-sample only a residual defined by subtracting the matching feature maps before and after traversing the levels $j+1, \ldots, n_{\text{levels}}$. By this definition, if the CNN-block has an identity mapping, then so does the whole MGIC-block in Alg. 1.

**Changing the channel resolution blocks**

The structure of the MGIC block as in Fig. 1 is more natural to equal input and output channel sizes, i.e., $c_{\text{in}} = c_{\text{out}}$. Hence, when we wish to change the number of channels , we define a lightweight shortcut that is designed to transform a tensor from $c_{\text{in}}$ to $c_{\text{out}}$ such that our MGIC-blocks will be given an input where $c_{\text{in}} = c_{\text{out}}$. Specifically, to obtain low computational cost, we use a depth-wise $3 \times 3$ convolution, although other alternatives such as a $1 \times 1$ convolution are also possible. In case we wish to change the spatial dimensions of the input tensor, we perform the same operation, only with a stride of 2.

**The complexity of the MGIC-block**

Consider a case where we have $c = c_{\text{in}} = c_{\text{out}}$ channels in the network, and we apply a standard convolution layer using $d \times d$ convolution kernels (e.g., a $3 \times 3$ kernel). The output consists of $c$ feature maps, where each one is a sum of the $c$ input maps, each convolved with a kernel. Hence, such a convolution layer requires $\mathcal{O}(c^2 \cdot d^2)$ parameters, inducing a quadratic growth in the parameters and FLOPs.

**Relaxation cost per level**

On each level of an MGIC block, a relaxation step is performed. At the $j$-th level, this relaxation step is realized by a grouped convolution of kernel size $d \times d$, with a group size of $s_g$ that divides $\frac{c}{2^j}$ (since at each level we halve the channels space, starting from $c$ channels), yielding $\frac{c}{s_g \cdot 2^j}$ groups.

Therefore, the number of parameters required for such relaxation step is $\frac{s_g \cdot c \cdot d^2}{2^j}$. At the coarsest level, we have $s_c$ channels which perform a fully-coupled relaxation step, requiring $s_c \cdot d^2$ parameters.

**The cost of restriction and prolongation**

As discussed in Sec. A, the restriction and prolongation operators are implemented via grouped $1 \times 1$ convolutions, halvening and doubling the feature space dimension, respectively. Those operators are learned at each level of our MGIC block. Therefore, the number of parameters for those operators at the $j$-th level is $\frac{c}{2^j}$. The analysis here is similar to the case of the relaxtion steps, only here $d = 1$, and we have no fully-coupled operators on the coarsest level.

**The total cost of an MGIC block**

Combining the analysis from the paragraphs above, the total number of parameters for an MGIC block with $n$ levels is as follows:

$$\sum_{j=0}^{n-1} \left( \frac{s_g \cdot c \cdot (d^2 + 1)}{2^j} \right) + s_c^2 \cdot d^2 < 2 \left( s_g \cdot c \cdot (d^2 + 1) \right) + s_c^2 \cdot d^2. \tag{7}$$

If $s_c$ is small (typically, we choose $s_c = s_g$) , we can neglect the term $s_c^2 \cdot d^2$ to obtain $\mathcal{O}(s_g \cdot c \cdot (d^2 + 1))$ parameters. Therefore, since $s_g$ and $s_c$ are fixed and small, and the spatial dimension of the learned relaxation step convolution kernel size $d$ is typically of small size (3, 5, or 7), our method scales linearly with respect to the network's width. This will be most beneficial if $c$ is large, which is typical in and usually required in order to obtain state-of-the-art performance on various tasks as in discussed in Sec. 1.

Table 4: Network architecture used in the implicit function representation experiment. – denotes a non-applicable parameter. $\alpha$ denotes a width-multiplier. BN denotes a batch-normalization operator. CNN-block can be any block (e.g., MGIC-block). $n$ denotes the number of points.

| Input | Operations | Expansion | $c_{out}$ |
|---|---|---|---|
| $n \times 2$ | $1 \times 1$ Conv, BN, ReLU | – | 16 |
| $n \times 16$ | CNN-block | 2 | $\alpha \cdot 160$ |
| $n \times \alpha \cdot 160$ | CNN-block | 2 | $\alpha \cdot 160$ |
| $n \times \alpha \cdot 160$ | $1 \times 1$ Conv, BN, ReLU | – | 64 |
| $n \times 64$ | $1 \times 1$ Conv, BN, ReLU | – | 2 |

**Memory footprint**

During training, the memory footprint of MGIC is roughly twice larger than a single CNN block since all the maps in the hierarchy are saved for backpropagation. However, during inference (which is more important here), the coarser feature maps are released while going up the hierarchy. When applying the upmost CNN block, the memory footprint is identical to a single block. Following the complexity analysis above, all the feature maps $\{\mathbf{x}^{(l)}\}$ require about $\times 2$ the memory of $\mathbf{x}^{(0)}$, but the memory footprint of a CNN block can be higher than that. For example, some MobileNets[16] involve an inverse bottleneck with $\times 6$ expansion rendering it more than three times as expensive as the additional MGIC overhead.

## B  Experiments

In this section we elaborate on the experiments in the main paper, and also report on additional experiments that are carried to verify the effectiveness of our MGIC method. We conduct an additional proof-of-concept experiment, measuring how good our MGIC is capable of approximating implicit functions. Then, we test our method on image classification and segmentation and point cloud classification benchmarks. Our goal is to compare how different architectures perform using a relatively small number of parameters, aiming to achieve similar or better results with fewer parameters and FLOPs. We train all our models using an NVIDIA Titan RTX and implement our code using the PyTorch software [32]. The details of the architectures that we use throughout this section are given in the supplementary material.

### B.1  Performance in function approximation

We study the capacity of our network by experimenting its efficacy in function approximation in a supervised learning setup. We claim that if a network can faithfully approximate a function $f(\vec{x})$ : $\mathbb{R}^n \to \mathbb{R}$, then its capacity, or representation power is high. This property is important, especially in applications where we wish to model implicit function via a neural networks, such as signed-distance fields for shape reconstruction and completion [31], and solution of PDEs [30, 33, 3, 26]—these works in particular approximate functions in an unsupervised manner. Here, we wish to approximate the function $f(x, y) = \cos(x) \sin(20y)$, given $(x, y) \in [0, 1]^2$. We use three networks – MobileNetV3 [16], GhostNet [12] and our MGIC. Since the input is a vector of two scalars (treated as channels of size 1), the convolution kernels are effectively only $1 \times 1$ convolutions. (Note that in any case these are the dominant operations in all the networks and are the driving force of neural network in general.) Throughout all the experiments, we used the network described in Tab. 4, where CNN-block is replaced with the respective method, with $\alpha = 0.6, 0.8, 1.2$. The settings of this experiments are as follows: we sample $50,000$ points from the surface of $f$ and train each network for $1,000$ epochs with a batch size of 128 points, using the SGD optimizer with a constant learning rate of $0.0001$. The loss function is the mean squared error (MSE). The results, summarized in Fig. 4 suggest that our MGIC has higher capacity, as it yields lower MSE with a lower number of parameters.

### B.2  Image classification

We compare our approach with a variety of popular and recent networks like ResNet-50 [13], MobileNetV3 [16] and GhostNet [12] for image classification on the CIFAR10 and ImageNet

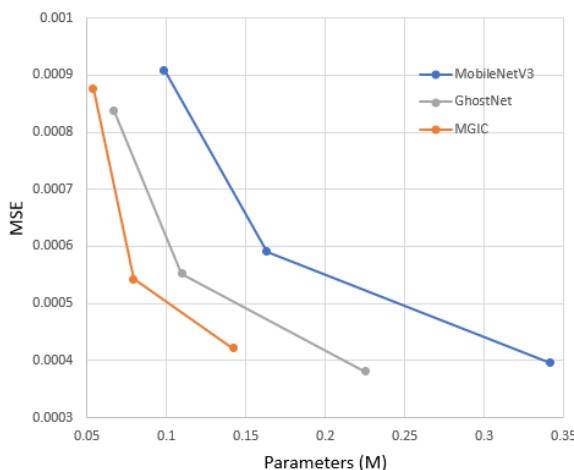

Figure 4: Reconstruction of $f(x, y) = \cos(x)\sin(20y)$ with MobileNetV3, GhostNet and our MGIC. Metric is in MSE as function of number of parameters.

datasets. We use SGD optimizer with a mini-batch size of 256 for ImageNet, and 128 for CIFAR-10, both for 100 epochs. Our loss function is cross-entropy. The initial learning rates for CIFAR-10 and ImageNet are 0.001 and 0.1, respectively. We divide them by 10 every 30 epochs. The weight decay is 0.0001, and the momentum is 0.9. As data augmentation, for both datasets, we use standard random horizontal flipping and crops, as in [13].

### B.2.1 CIFAR-10

The CIFAR-10 dataset [21] consists of 60K natural images of size $32 \times 32$ with labels assigning each image into one of ten categories. The data is split into 50K training and 10K test sets. Here, we use a ResNet-56 [13] architecture together with our MGIC block, with parameters $s_g = 8$, $s_c = 16$. We compare our method with other recent and popular architectures such as AMC-ResNet-56 [14] and Ghost-ResNet-56 [12], and our baseline is the original ResNet-56. We report our results in Tab. 5, where we see large improvement over existing methods, while retaining low number of parameters and FLOPs.

Table 5: Comparison of state-of-the-art methods for compressing ResNet-56 on CIFAR-10. - indicates unavailable results.

| Architecture | Params [M] | FLOPs [M] | Test acc. |
|---|---|---|---|
| ResNet-56 [13] | 0.85 | 125 | 93.0% |
| CP-ResNet-56 [15] | - | 63 | 92.0% |
| $\ell_1$ -ResNet-56 [24] | 0.73 | 91 | 92.5% |
| AMC-ResNet-56 [14] | - | 63 | 91.9% |
| Ghost-ResNet-56 [12] | 0.43 | 63 | 92.7% |
| **MGIC-ResNet-56 (ours)** | 0.41 | 60 | **94.2%** |

### B.2.2 ImageNet

**ResNet-50 compression**

We compress the ResNet-50 architecture, and compare our MGIC approach with other methods. As the goal of this experiment is to compress a standard ResNet-50 [13], we follow the exact architecture of the latter, only replacing each ResNet block layer by an MGIC-ResNet block, denoted as MGIC(·), as depicted in Tab. 6. The results are reported in Tab. 7, where we propose three variants of our MGIC-ResNet-50 network, differing in the $s_g$ and $s_c$ parameters. Our results outperform the rest of the considered methods, and our network with $s_g = 64$, $s_c = 64$ also outperforms ResNeXt-50 [44]

Table 6: MGIC-ResNet50 architecture. MGIC $(\cdot)$ is the MGIC version of the given block. Conv2D is a 2D convolution layer followed by a BatchNorm operation and a ReLU non-linear activation. # Rep is the number of block repetitions. $c_{out}$ denotes the number of output channels. A maxpool operation occurs after every convolution and MGIC layer.

| Input | Layer | # Rep | $c_{out}$ |
|---|---|---|---|
| $224^2 \times 3$ | Conv2D $7 \times 7$ | 1 | 64 |
| $112^2 \times 64$ | Conv2D $3 \times 3$ | 1 | 64 |
| $112^2 \times 64$ | MGIC $\left( \begin{bmatrix} 1 \times 1 , 64 \\ 3 \times 3 , 64 \\ 1 \times 1 , 256 \end{bmatrix} \right)$ | 3 | 256 |
| $56^2 \times 256$ | MGIC $\left( \begin{bmatrix} 1 \times 1 , 128 \\ 3 \times 3 , 128 \\ 1 \times 1 , 512 \end{bmatrix} \right)$ | 4 | 512 |
| $28^2 \times 512$ | MGIC $\left( \begin{bmatrix} 1 \times 1 , 256 \\ 3 \times 3 , 256 \\ 1 \times 1 , 1024 \end{bmatrix} \right)$ | 6 | 1024 |
| $14^2 \times 1024$ | MGIC $\left( \begin{bmatrix} 1 \times 1 , 512 \\ 3 \times 3 , 512 \\ 1 \times 1 , 2048 \end{bmatrix} \right)$ | 3 | 2048 |
| $7^2 \times 2048$ | AvgPool2D $7 \times 7$ | 1 | 2048 |
| $1^2 \times 2048$ | FC | 1 | 1000 |

Table 7: Comparison of state-of-the-art methods for compressing ResNet-50 on ImageNet dataset.

| Model | Params [M] | FLOPs [B] | Top-1 Acc.% | Top-5 Acc.% |
|---|---|---|---|---|
| ResNet-50 [13] | 25.6 | 4.1 | 75.3 | 92.2 |
| Thinet-ResNet-50 [27] | 16.9 | 2.6 | 72.1 | 90.3 |
| NISP-ResNet-50-B [47] | 14.4 | 2.3 | - | 90.8 |
| Versatile-ResNet-50 [40] | 11.0 | 3.0 | 74.5 | 91.8 |
| SSS-ResNet-50 [19] | - | 2.8 | 74.2 | 91.9 |
| Ghost-ResNet-50 [12] | 13.0 | 2.2 | 75.0 | 92.3 |
| MGIC-ResNet-50 ($s_g = 32$, $s_c = 64$) (ours) | 9.4 | 1.6 | 75.8 | 92.9 |
| MGIC-ResNet-50 ($s_g = 64$, $s_c = 64$) (ours) | 15.1 | 2.5 | **77.9** | **93.7** |
| Shift-ResNet-50 [42] | 6.0 | - | 70.6 | 90.1 |
| Taylor-FO-BN-ResNet-50 [29] | 7.9 | 1.3 | 71.7 | - |
| Slimmable-ResNet-50 0.5× [46] | 6.9 | 1.1 | 72.1 | - |
| MetaPruning-ResNet-50 [25] | - | 1.0 | 73.4 | - |
| Ghost-ResNet-50 (s=4) [12] | 6.5 | 1.2 | 74.1 | 91.9 |
| MGIC-ResNet-50 ($s_g = 16$, $s_c = 64$) (ours) | 6.2 | 1.0 | **74.3** | **92.0** |

(25.0M parameters, 4.2B FLOPs, 77.8% top-1 accuracy), which is not shown in the table because the ResNeXt architecture utilizes more channels than ResNet-50 and therefore is not directly comparable.

**Image classification on a budget of FLOPs**

In this experiment we compare our approach with recent light networks. In particular, we follow the MobileNetV3-Large [16] architecture for its efficiency and high accuracy, and replace the standard MobileNetV3 block with our MGIC block. Our building blocks are MGIC-Bottlenecks (dubbed MGIC-bneck). That is, we build a MGIC version of the bottleneck from MobileNetV3. Our MGIC-MobileNetV3 is given in Tab. 9. Note, this is the ×1.0 version, and can be modified via the width multiplier $\alpha$. Our parameter $s_g$ controls the group size – therefore it determines the number of groups in each MGIC bottleneck. We denote the number of output channels by $c_{out}$ and the number of hidden channels within a block (the dimension of the square operator $K_{l_2}$ in a the block (**??**)), also

Table 8: Comparison of state-of-the-art light-weight networks on ImageNet dataset classification.

| Model | Params [M] | FLOPs [M] | Top-1 Acc.% | Top-5 Acc.% |
|---|---|---|---|---|
| ShuffleNetV1 0.5× (g=8) [49] | 1.0 | 40 | 58.8 | 81.0 |
| MobileNetV2 0.35× [34] | 1.7 | 59 | 60.3 | 82.9 |
| ShuffleNetV2 0.5× [28] | 1.4 | 41 | 61.1 | 82.6 |
| MobileNeXt 0.35× [50] | 1.8 | 80 | 64.7 | - |
| MobileNetV3-Small 0.75× [16] | 2.4 | 44 | 65.4 | - |
| GhostNet 0.5× [12] | 2.6 | 42 | 66.2 | 86.6 |
| MGIC-MobileNetV3 0.6× (ours) | 2.3 | 48 | **67.0** | **87.3** |
| MGIC-MobileNetV3 0.6× (ours) no h–swish | 2.3 | 45 | **66.8** | **86.9** |
| MobileNetV1 0.5× [17] | 1.3 | 150 | 63.3 | 84.9 |
| MobileNetV2 0.6× [34] | 2.2 | 141 | 66.7 | - |
| ShuffleNetV1 1.0× (g=3) [49] | 1.9 | 138 | 67.8 | 87.7 |
| ShuffleNetV2 1.0× [28] | 2.3 | 146 | 69.4 | 88.9 |
| MobileNeXt 0.75× [50] | 2.5 | 210 | 72.0 | - |
| MobileNetV3-Large 0.75× [16] | 4.0 | 155 | 73.3 | - |
| GhostNet 1.0× [12] | 5.2 | 141 | 73.9 | 91.4 |
| MGIC-MobileNetV3 1.0× (ours) | 5.2 | 145 | **74.8** | **92.0** |
| MGIC-MobileNetV3 1.0× (ours) no h–swish | 5.2 | 138 | **74.3** | **91.6** |
| MobileNetV2 1.0× [34] | 3.5 | 300 | 71.8 | 91.0 |
| ShuffleNetV2 1.5× [28] | 3.5 | 299 | 72.6 | 90.6 |
| FE-Net 1.0× [7] | 3.7 | 301 | 72.9 | - |
| FBNet-B [41] | 4.5 | 295 | 74.1 | - |
| ProxylessNAS [5] | 4.1 | 320 | 74.6 | 92.2 |
| MnasNet-A1 [36] | 3.9 | 312 | 75.2 | 92.5 |
| MobileNeXt 1.0× [50] | 3.4 | 300 | 74.0 | - |
| MobileNetV3-Large [16] 1.0× | 5.4 | 219 | 75.2 | - |
| GhostNet 1.3× [12] | 7.3 | 226 | 75.7 | 92.7 |
| MGIC-MobileNetV3 1.2× (ours) | 7.1 | 233 | **76.1** | **93.2** |
| MGIC-MobileNetV3 1.2× (ours) no h–swish | 7.1 | 217 | **76.2** | **93.4** |

referred to as the expansion size, by $\#exp$. In case $c_{out}$ and $\#exp$ are not divisible by $s_g$, we set $s_g$ to the closest (smaller) integer to its intended value such that it divides them. For example, in our experiments we set $s_g = 64$, and for the network defined in Tab. 9, the 5th MGIC-bneck layer has $\#exp = 120$ and $c_{out} = 40$, meaning they do not divide by 64. Therefore we modify $s_g$ to be the largest integer that is smaller than 64 and divides both $\#exp$ and $c_{out}$, giving $s_g = 40$ in this example. Our experiment is divided into three scales - small, medium, and large, where we scale our networks with width factors of $0.6$, $1.0$ and $1.2$, respectively. We find that our method obtains higher accuracy, with a similar number of FLOPs, as depicted from the results in Tab. 8 and Fig. 3. Specifically, we compare our methods with and without the use of the h–swish activation function [16], where we see similar results. Compared to other popular and recent methods like MobileNetV3, GhostNet and ShuffleNetV2, we obtain better accuracy given the same FLOPs.

**Inference times.** We measure the single thread inference times on one image using lightweight models on a Samsung Galaxy S8 mobile device (using the TFLite tool [1]), and an Intel i9-9820X CPU—see Tab. 10 (averaged over 50 inferences). We observe that at least by these timings, the runtime of MGIC is on par with the considered architectures while obtaining higher accuracy.

### B.3 Image semantic segmentation

We compare our method with MobileNetV3 on semantic segmentation on the Cityscapes [8] dataset. For the encoder part of the network, we build large and small variants, based on MobileNetV3-Large and MobileNetV3-Small, described in Tables 1-2 in [16], respectively. We also utilize the same LR-ASPP segmentation head and follow the observations from [16]. Namely, we reduce the number of channels in the last block of our networks by a factor of two and use 128 filters in the segmentation

Table 9: MGIC-MobileNetV3 architecture. MGIC-bneck denotes a MGIC-Bottleneck . The bottleneck is the same as in MobileNetV3, only in a multigrid-in-channels form. Conv2D is a 2D convolution layer followed by a BatchNorm operation and a ReLU non-linear activation. # exp denotes the expansion size. $c_{out}$ denotes the number of output channels. SE stands for Squeeze-Excite. Pool denotes a maxpool operation, reducing the spatial size of the input . - denotes a non-applicable option. ✓ and × denote True and False, respectively.

| Input | Operation | # exp | $c_{out}$ | SE | Pool |
|---|---|---|---|---|---|
| $224^2 \times 3$ | Conv2D $3 \times 3$ | 16 | - | × | ✓ |
| $112^2 \times 16$ | MGIC-bneck | 16 | 16 | × | × |
| $112^2 \times 16$ | MGIC-bneck | 48 | 24 | × | ✓ |
| $56^2 \times 24$ | MGIC-bneck | 72 | 24 | × | × |
| $56^2 \times 24$ | MGIC-bneck | 72 | 40 | ✓ | ✓ |
| $28^2 \times 40$ | MGIC-bneck | 120 | 40 | ✓ | × |
| $28^2 \times 40$ | MGIC-bneck | 240 | 80 | × | ✓ |
| $14^2 \times 80$ | MGIC-bneck | 200 | 80 | × | × |
| $14^2 \times 80$ | MGIC-bneck | 184 | 80 | × | × |
| $14^2 \times 80$ | MGIC-bneck | 184 | 80 | × | × |
| $14^2 \times 80$ | MGIC-bneck | 480 | 112 | ✓ | × |
| $14^2 \times 112$ | MGIC-bneck | 672 | 112 | ✓ | × |
| $14^2 \times 112$ | MGIC-bneck | 672 | 160 | ✓ | ✓ |
| $7^2 \times 160$ | MGIC-bneck | 960 | 160 | × | × |
| $7^2 \times 160$ | MGIC-bneck | 960 | 160 | ✓ | × |
| $7^2 \times 160$ | MGIC-bneck | 960 | 160 | × | × |
| $7^2 \times 160$ | MGIC-bneck | 960 | 160 | ✓ | × |
| $7^2 \times 160$ | Conv2D $1 \times 1$ | - | 960 | × | × |
| $7^2 \times 960$ | AvgPool $7 \times 7$ | - | 960 | × | × |
| $1^2 \times 960$ | Conv2D $1 \times 1$ | - | 1280 | × | × |
| $1^2 \times 1280$ | FC | - | 1000 | × | × |

Table 10: Inference runtime of state-of-the-art small networks on a Samsung Galaxy S8 mobile device and a PC CPU

| Metric | MobileNetV2 1.0x | MobileNetV3 0.75x | GhostNet 1.0x | MGIC MobileNetV3 1.0x |
|---|---|---|---|---|
| Accuracy [%] | 71.8 | 73.3 | 73.9 | 74.8 |
| Mobile runtime [ms] | 795 | 418 | 487 | 480 |
| PC runtime [ms] | 130 | 140 | 170 | 172 |

head. For training, we use the same data augmentation and optimization approach as in [6]. The results are shown in Tab. 11, where report the mean intersection over union (mIoU) metric of our MGIC-Large with $s_g = 64$ and $s_g = 32$. We note that the results for the former are slightly better than those of MobileNetV3, while the performance of the latter are more favorble as they offer similar accuracy for less FLOPs and parameters. In addition, we read similar accuracy when using our MGIC-Small with $s_g = 64$.

## B.4  Point cloud classification

We define the architecture in Tab. 12, where G-conv denotes a graph convolution layer, according to the methods listed in Tab. 3, followed by a BatchNorm operation and a ReLU non-linear activation. MLP is realized by a simple $1 \times 1$ convolution followed by a BatchNorm operation and a ReLU non-linear activation. FC is a fully connected layer. In all networks, we define the adjacency matrix using the k-NN algorithm with $k = 10$ .Then, we replace the GCN block with each of the backbones

Table 11: Segmentation results on Cityscapes dataset. Metric is in mean intersection over union.

| Backbone | Params[M] | FLOPs [B] | mIoU % |
|---|---|---|---|
| MobileNetV3-Large | 1.51 | 9.74 | 72.64 |
| MobileNetV3-Small | 0.47 | 2.90 | 68.38 |
| MGIC-Large $s_g = 64$ (ours) | 1.67 | 9.62 | 72.69 |
| MGIC-Large $s_g = 32$ (ours) | 1.32 | 8.87 | 71.02 |
| MGIC-Small $s_g = 64$ (ours) | 0.48 | 2.73 | 68.52 |

listed in Tab. 3, where we also report their performance on point-cloud classification on ModelNet-10 [43] benchmark where we sample $1,024$ points from each shape.

Table 12: Graph neural network for point cloud classification. G-conv is a graph convolution layer. MLP is a multi-layer perceptron. MaxPool is a global max-pooling layer. $c_{out}$ denotes the number of output channels. $\times$ denotes the number of repetitions of the respective layer.

| Input | Operation | $c_{out}$ |
|---|---|---|
| $1024 \times 3$ | G-conv | 64 |
| $1024 \times 64$ | $2 \times$ G-conv | 64 |
| $1024 \times 64$ | MLP | 64 |
| $1024 \times 64$ | $3 \times$ G-conv | 64 |
| $1024 \times 64$ | MLP | 64 |
| $1024 \times 64$ | MaxPool | 64 |
| $1 \times 64$ | $3 \times$ MLP | 64 |
| $1 \times 64$ | FC | 10 |

## B.5   Ablation study

To determine the impact of the parameters $s_g$ and $s_c$, we experiment on CIFAR-10 for image classification. First, we fix $s_g$ to 16 and observe how the number of parameters, FLOPs, and accuracy of MGIC-ResNet-56 change. Secondly, we fix $s_c$ to 16, while modifying $s_g$, and examine our model's behavior. Our conclusion from the results reported in Tab. 13, is that a growth in $s_g$ or $s_c$ yields better accuracy at the cost of more parameters and flops, since an increased communication between the channels is allowed. However, we also note that in this experiment, an increase of $s_c$ past 16 or $s_g$ past 8 does not yield significant accuracy improvement. Thus, it suggests that our method is capable of faithfully reducing the feature space given a budget of parameters or FLOPs.

Table 13: Influence of $s_c$ and $s_g$ in our MGIC framework on ResNet-56 architecture and CIFAR-10 dataset. $s_g$ is fixed to 16.

| $s_c$ | $s_g$ | Params[M] | FLOPs [M] | Accuracy % |
|---|---|---|---|---|
| 64 | 16 | 0.53 | 91 | 94.7 |
| 32 | 16 | 0.5 | 76 | 94.6 |
| 16 | 16 | 0.47 | 65 | 94.3 |
| 16 | 32 | 0.79 | 100 | 94.8 |
| 16 | 16 | 0.53 | 85 | 94.7 |
| 16 | 8 | 0.41 | 60 | 94.2 |
| 16 | 4 | 0.29 | 45 | 92.8 |

