# OpenReview forum: "MGIC: Multigrid-in-Channels Neural Network Architectures"
_NeurIPS.cc/2021/Workshop/DLDE — DLDE Workshop -- NeurIPS 2021 Poster_

### Official Review · Reviewer_Ni1d · 2021-10-02
**Significant improvement for CNN**

**Confidence:** 4

**Review:**

As the abstract of this paper declares,
this work reduces the channel complexity of CNN by applying
multigrid methods. The quadratic complexity is reduced to linearity.

This article is easy to read even for non-exports of CNN and multi-grid methods.


I think the second and third paragraph of this article have some overlap
and can be merged together. The two paragraphs focus on the existing
challenges but do not tell clearly the related techniques to solve
the complexity problems of CNN. This part is also lacking in
the experiments, which can be regarded as one shortcoming of this paper.


**Score:**

4: Very good paper

---

### Official Review · Reviewer_5Kxe · 2021-10-03

**Confidence:** 3

**Review:**

This work investigates empirical effectiveness of a multigrid CNN variant.

**Novelty & Significance**

The paper tackles the important problem of inference on a computational budget. While preliminary experiments show promising results against common baselines, the authors do not provide comparisons against other multigrid CNN variants (of which there are many). It is thus impossible to determine if the gains are due to their specific proposal or due to experiment design.

**Questions**:
* The authors mention an increased memory footprint for MGIC. What is the footprint for MGIC on ImageNet (with the given batch size)? Can the authors provide an asymptotic estimate of its scaling, much like they've done with parameter numbers?


**Score:**

3: Good paper

---

### Official Review · Reviewer_YT1h · 2021-10-10
**A thoughtful insight about how to reduce redundant connections in CNN**

**Confidence:** 3

**Review:**

This work presents a promising variant of fully connected CNN layer that reduce the quadratic growth of parameters to a linear one. The growth of parameters in conventional FC-CNN could bring down the marginal benefit of more parameters due to the difficulty in training. This work points out a potential direction, to which the deep FC-CNN can be reduced to and represented by much lighter hierarchical structures that behave almost equivalently on the empirical data distribution, as shown in the brief results presented in this work.

Since this is only an extended abstract with fairly limited length, I would say it would be interesting to see in the poster presentation, about how the MGIC differs from the existing grouped convolution. And I think it would also be interesting to see what the advantages of including the FC-CNN in the coarsest level, as opposed to the other grouped convolution methods that do not have such structure.

Pros:
- Thoughtful design for the lighter hierarchical structure to partially retain correlation between channels.
- A range of solid results comparing to other networks under different scenarios.

Cons
- The authors did not include comparison and contrast to previous work, which makes readers from other field difficult to assess the novelty (but understandable since it is just a 4-pages long extended abstract.)
- The terminology is a bit confusing for the description of the architecture.

**Score:**

4: Very good paper

---

### Official Review · Reviewer_RrZo · 2021-10-11
**Unclear relevance to the DLDE workshop**

**Confidence:** 4

**Review:**

This paper proposes a mechanism of replacing convolutional blocks in standard experiments with a different block called "MGIC-block". Although the authors have given extensive experimental data in support of the improvements seen, the actual operation done by MGIC-block remains unclear in the absence of a mathematical expression. The paper spends too much time on displaying experimental data of questionable relevance while no importance is given to giving explicit mathematical expressions for what function does the MGIC-block compute.

But the main issue here is that I see no connection between the study in this paper and the theme of this workshop on the intersection of deep-learning and differential equations. This looks like a misplaced submission!


**Score:**

1: Reject: trivial or wrong

---

### Decision · Program_Chairs · 2021-10-16

**Decision:**

Accept (Poster)

**Comment:**

The reviewers seem to have mostly appreciated the content of the article, even if not clearly aligned with the topic of the Workshop.